# Chester: A Web Delivered Locally Computed Chest X-ray Disease Prediction System

**Author(s) names withheld**                                                   EMAIL(S) WITHHELD

## Abstract

In order to bridge the gap between Deep Learning researchers and medical professionals we develop a very accessible free prototype system which can be used by medical professionals to understand the reality of Deep Learning tools for chest X-ray diagnostics. The system is designed to be a second opinion where a user can process an image to confirm or aid in their diagnosis. Code and network weights are delivered via a URL to a web browser (including cell phones) but the patient data remains on the users machine and all processing occurs locally. This paper discusses the three main components in detail: out-of-distribution detection, disease prediction, and prediction explanation. The system open source and freely available here: REMOVED

**Keywords:** Chest X-ray, Radiology, Deep Learning, JavaScript

## 1. Introduction

Deep learning tools for medicine hold promise to improve the standard of medical care. However this technology is still in the research phase and is not being adopted fast enough due to uncertainty over business models, lack of understanding of Artificial Intelligence (AI) technology in hospitals, limited expertise in health data science and artificial intelligence technologies, hospital inertia, data access issues, and regulatory approval hurdles. These issues are preventing the necessary collaboration between AI researchers and the doctors who will use these tools.

Our approach is to build a very accessible prototype system which can be used by medical professionals to understand the reality of Deep Learning tools for chest X-ray diagnostics. The system is designed to be a second opinion where a user can process an image to confirm or aid in their diagnosis. This paper describes a web based (but locally run) system for diagnostic prediction of chest X-rays. Code is delivered via a URL[1] where users are presented an interface shown in Figure 1. This approach has many advantages. First, the patient data remains on the users machine in order to preserve privacy. Second, all processing occurs locally allowing us to scale the computation at minimum cost. Other strategies for software deployment, such as shipping a desktop application, incur a larger development overhead which would be unsustainable for a free prototype. This would also introduce difficulty with hospital IT departments who are gatekeepers for the software that doctors use yet largely allow access to websites.

---

1. REMOVED

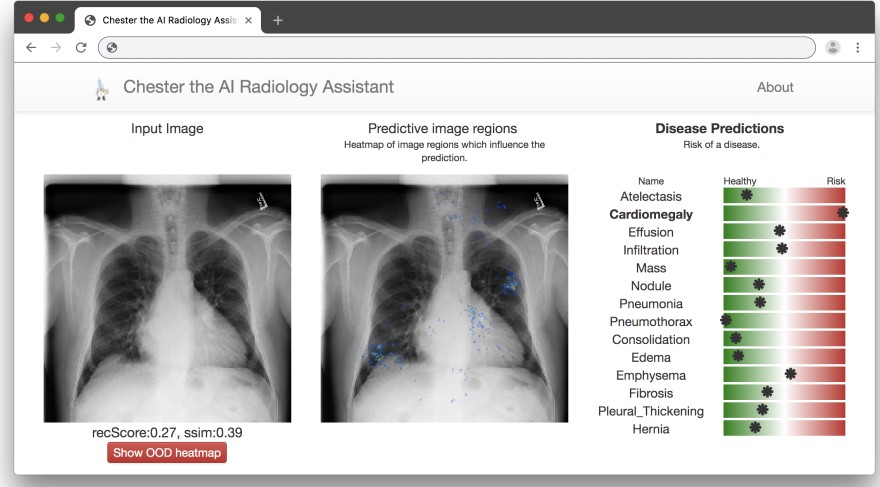

Figure 1: The web interface is designed to be simple tool which provides diagnostic information as well as a visual explanation of the prediction.

The goals of this tool are as follows:

1. Enabling the medical community to experiment with Deep Learning tools to identify out how they can be useful and where they fail; providing feedback so the machine learning community can identify challenges to work on.

2. To serve as an example of what is possible when data is open. It only exists because the NIH released 100k images (Wang et al., 2017) for unrestricted use that were used to train the model.

3. Across the entire world we can establish a lower bound of care. Every radiologist with a web browser should be no worse than this tool. We can transfer the knowledge of the NIH to everyone.

4. The design of the system is a model that can be copied to globally scale medical solutions without incurring significant server costs. No need to configure a computer with deep learning libraries.

The system is composed of three main parts: out of distribution (OOD) detection, diagnostic prediction, and prediction explanation. Together these parts form a complete tool which can be used to complement the skills of a radiologist, doctor, or student. In developing this system we have identified many challenges which we analysed solutions for in this work.

One difficult challenge is knowing when to not make a prediction (the focus of §3). If you asked a radiologist to diagnose an image of something that is not their speciality they should not provide a diagnosis. For neural networks their *speciality* is the domain of the classifier which is defined by their training distribution. We cannot evaluate the accuracy of the model on these examples so we choose not to process them. This is illustrated in Figure 2.

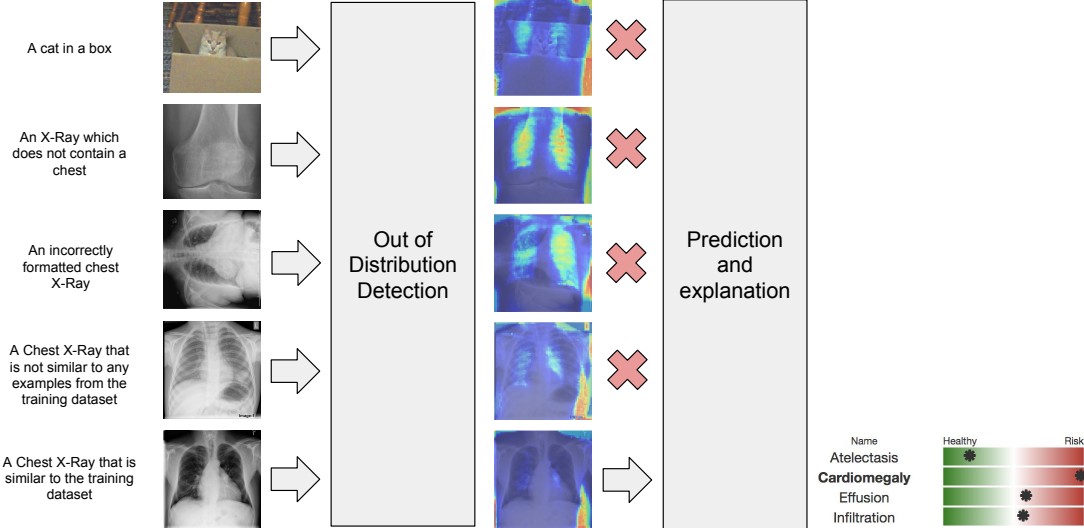

Figure 2: The use cases of an out-of-distribution detection method. Using a reconstruction loss we can visualize where the error occurs on the image. Only images which are similar to those in the training distribution are allowed to be processed.

Explaining the prediction is critical to provide confidence that the model is correct as well as allowing users to come to their own conclusion with insight from the tool (focus of §4). Although many methods exist we have a limited computational budget.

Computing locally in the browser was the most technical challenge (focus of §5). With the recent creation of tools such as ONNX (Bai et al., 2017) and TensorFlow.js (Abadi et al., 2015) models trained in PyTorch (Paszke et al., 2017) can be converted to work in the browser and compute using WebGL.

## 2. Disease Prediction

We used a DenseNet-121 architecture (Huang et al., 2017) which was shown to work well on chest X-rays in Rajpurkar et al. (2017). It is trained with the train-validation-test split as in the initial paper (70%, 10%, 20%). To save space the data used is described in Appendix §A. We also compare our model with weights from Weng et al. (2017), which achieve comparable performance compared to the original paper as shown in Table 1.

**Data augmentation** We would like to assess, and possibly improve, the generalization capacity of our model. To do so we investigated whether data augmentation could help increase the size of the training domain of the model without hurting its performance, especially on the original domain (without data augmentation). We applied random rotation, translation, and scaling to the original images at both training and testing times. We use the PadChest (Bustos et al., 2019) dataset from Spain as an external validation dataset.

A notable subset of the results are shown in Figure 3 (with the full analysis in Appendix Figure E.2). The DenseNet achieves reasonably equivalent performance on the whole extended domain (with the highest level of data augmentation) without hurting performance on the original domain (without data augmentation). On the external dataset we observe good generalization. However, some classes are consistently predicted (across models) better or worse on the PadChest dataset. We note the surprising result of Infiltration is predicted

Table 1: Performance (AUC) of models on the ChestX-ray14 dataset. We include results from the Rajpurkar et al. (2017) and compute performance of our own model. Our standard deviation is computed using 10 random splits of the test set. Each split is half the size of the test set.

| Disease | ChestX-ray14 Wang et al. (2017) DenseNet-50 | CheXNet Rajpurkar et al. (2017) DenseNet-121 | CheXNet Py3 Weng et al. (2017) DenseNet-121 | CheXNet Ours 45d rot 15%trans/15%scale DenseNet-121 |
|---|---|---|---|---|
| Atelectasis | 0.71 | 0.80 | 0.81 ± 0.01 | 0.84 ± 0.01 |
| Cardiomegaly | 0.80 | 0.92 | 0.90 ± 0.01 | 0.92 ± 0.01 |
| Effusion | 0.78 | 0.86 | 0.87 ± 0.01 | 0.88 ± 0.01 |
| Infiltration | 0.60 | 0.73 | 0.70 ± 0.01 | 0.73 ± 0.01 |
| Mass | 0.70 | 0.86 | 0.82 ± 0.01 | 0.87 ± 0.01 |
| Nodule | 0.67 | 0.78 | 0.74 ± 0.01 | 0.79 ± 0.01 |
| Pneumonia | 0.63 | 0.76 | 0.76 ± 0.02 | 0.72 ± 0.04 |
| Pneumothorax | 0.80 | 0.88 | 0.83 ± 0.01 | 0.86 ± 0.01 |
| Consolidation | 0.70 | 0.79 | 0.79 ± 0.01 | 0.81 ± 0.01 |
| Edema | 0.83 | 0.88 | 0.86 ± 0.01 | 0.91 ± 0.01 |
| Emphysema | 0.81 | 0.93 | 0.89 ± 0.01 | 0.93 ± 0.01 |
| Fibrosis | 0.76 | 0.80 | 0.78 ± 0.01 | 0.78 ± 0.01 |
| Pleural Thickening | 0.70 | 0.80 | 0.75 ± 0.01 | 0.81 ± 0.01 |
| Hernia | 0.76 | 0.91 | 0.88 ± 0.03 | 0.83 ± 0.07 |

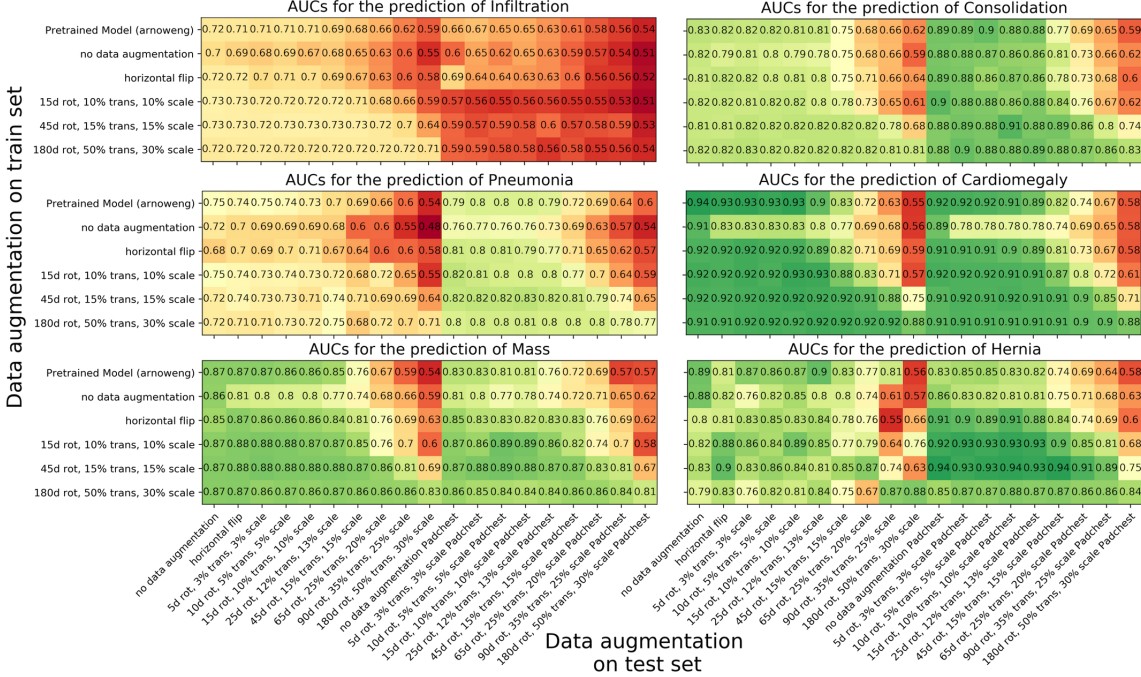

Figure 3: Robustness of the different models to data augmentation on the test set of the Chest Xray dataset (left half of matrices) and on the PadChest dataset (right half of matrices). There is an increasing level of data augmentation at train time (from top to bottom) and at test time (from left to right). This plot for all tasks is shown in Appendix Figure E.2

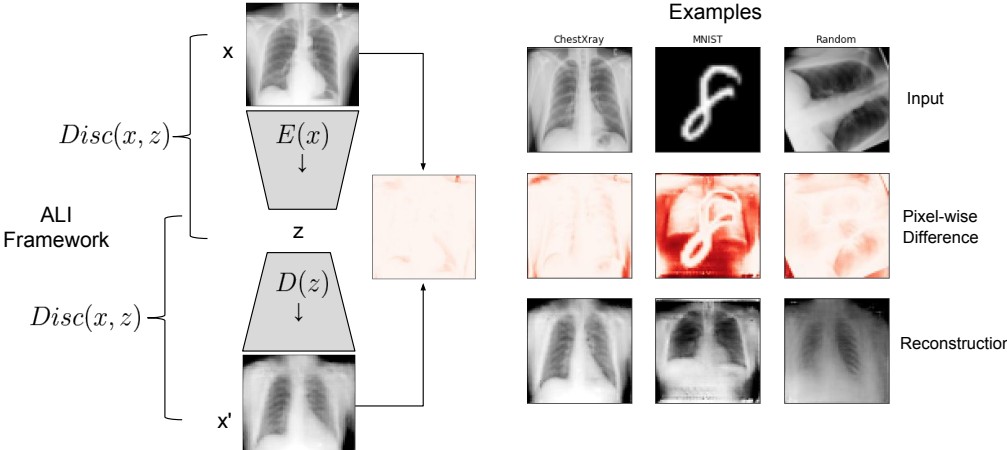

Figure 4: Example reconstructions using the ALI model. The pixel-wise error is shown in red.

worse while Consolidation is predicted better. One possible reason for this is inconsistencies between the annotations of the two datasets.

**Postprocessing** In order to calibrate the output probability, we apply a piecewise linear transformation Eq. 1 to the predictions so that the best operating point corresponds to a 50% disease probability while ensuring that predictive probabilities lie in the set $[0, 1]$. The ROC curves and chosen operating points are shown in Appendix Figure E.4. For each disease, we computed the optimal operating point by maximizing the difference (*True positive rate - False positive rate*).

$$f_{opt}(x) = \begin{cases} \frac{x}{2opt} & x \leq opt \\ 1 - \frac{1-x}{2(1-opt)} & otherwise \end{cases} \tag{1}$$

## 3. Predicting out of distribution (OoD)

We would like a function that will produce a score indicating how out of the training distribution a sample is. Shafaei et al. (2018) and Zenati et al. (2018a) analyzed many different approaches to this problem finding that density/variational/energy approaches don't work as well as reconstruction or using models trained for a classification task. Zenati et al. (2018b) found that using an approach with an encoder such as ALI/BiGAN (Dumoulin et al., 2016; Donahue et al., 2017) with reconstruction loss performed well.

The architecture for an autoencoder (AE) trained with ALI is shown in Figure 4. A discriminator ($Disc$) is used to pull the joint distributions between the encoder and decoder together. The encoder output is trained to conform to a Gaussian $E(\text{img}) \sim N(\mu, \sigma)$ (training details in Appendix §C). This approach produces chest X-ray looking reconstructions unlike an AE trained with an L2 loss. This allows the reconstruction error to indicate the regions of the image which are unlike a chest X-ray.

**Outlier Metrics** We experimented with multiple metrics to rank outlier images. First we try to use the latent variable of the ALI model because this variable should be a Gaussian

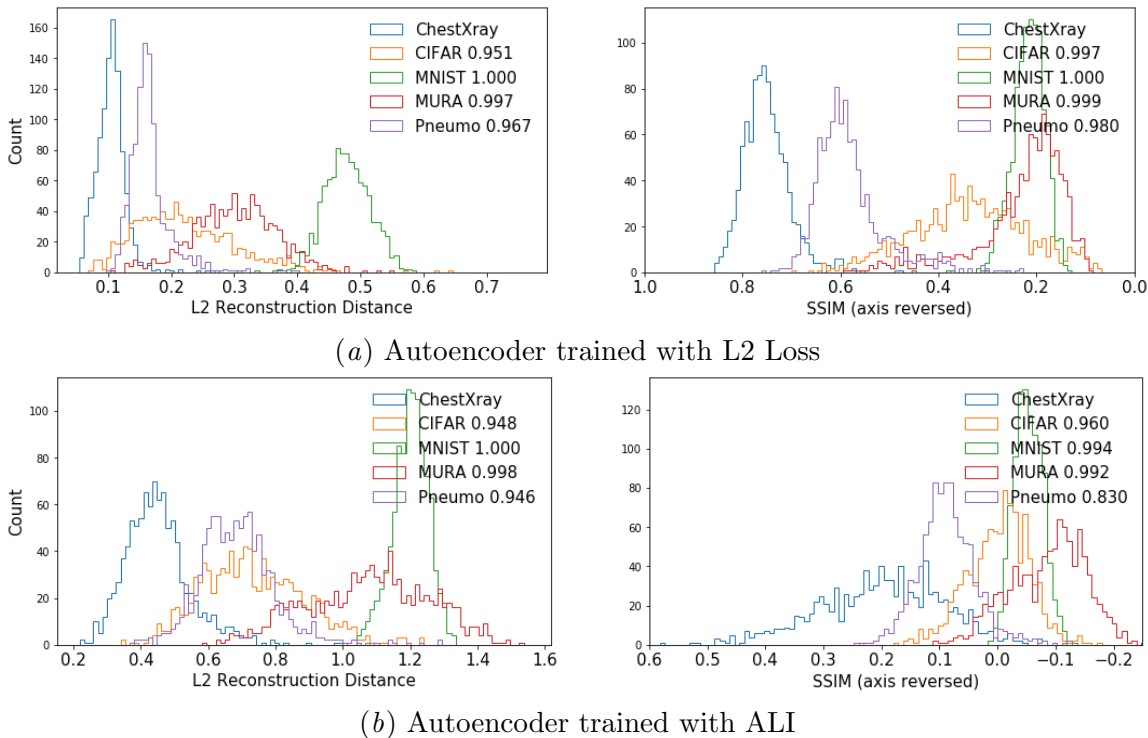

(a) Autoencoder trained with L2 Loss

(b) Autoencoder trained with ALI

Figure 5: Distribution of OoD distances using different metrics. Euclidean distance in the latent space and three types of pixel-wise reconstruction losses: L1, L2, and SSIM. The AUC when separating the ChestXray from each other dataset is shown in the legend.

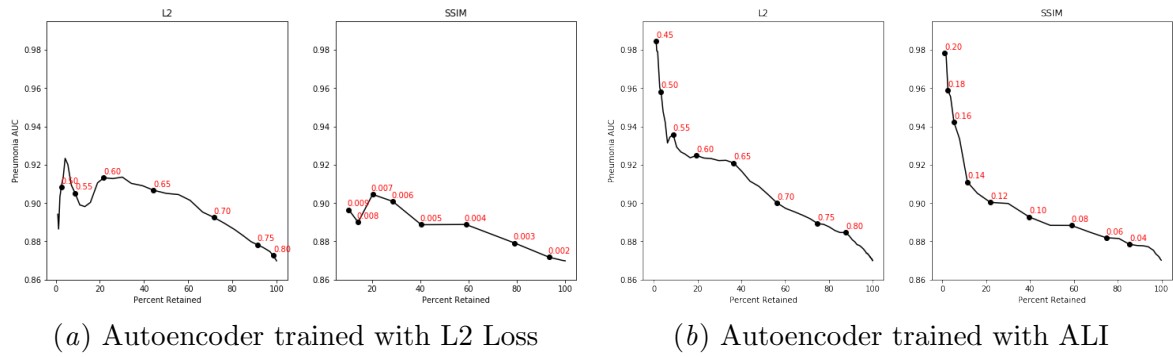

(a) Autoencoder trained with L2 Loss     (b) Autoencoder trained with ALI

Figure 6: The DenseNet-121 performance on Pneumonia dataset increase with stringency of outlier cutoff value (shown in red) while the percent of retained images decrease

centered at 0. We suspected that we could use the metric from the mean as an estimate of how likely the sample is in the training distribution. We also explored reconstruction error using L1, L2, and SSIM (Wang et al., 2004) metric between the original and reconstructed images. In Figure 5 we show a comparison between L2 and SSIM metrics for the AE and

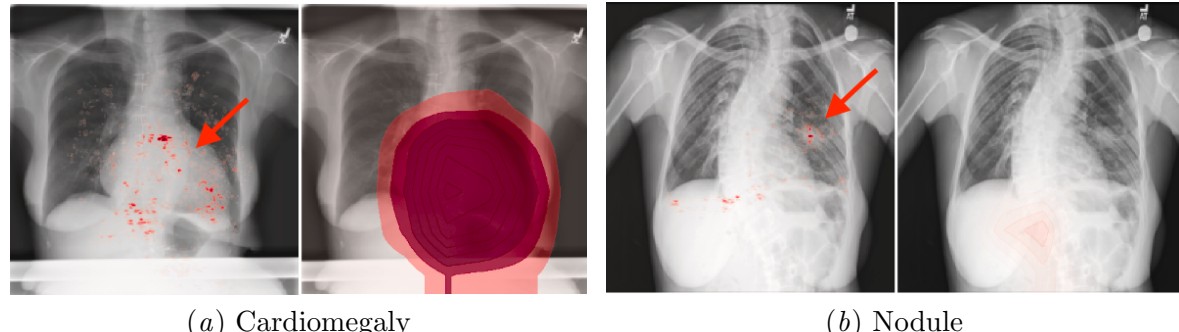

$(a)$ Cardiomegaly                    $(b)$ Nodule

Figure 7: Example of localization using the two different mappings on two images. In each subfigure Left: Saliency Map. Right: Class Activation Map. The more red a region is the more of an impact of that region. Transparent pixels have negligible impact on the prediction.

ALI model. L2 seems to confuse an external Pneumonia chest X-ray dataset (described in Appendix §A) with CIFAR natural images while the SSIM score is able to separate them.

We observed the latent variable distance method failed to identify most of the outliers, while the L1, L2, and SSIM metric gave similar performance on the CIFAR, MNIST and MURA datasets, shown in Appendix Figures E.1 and E.3. We observed that the CIFAR dataset is the most challenging dataset. This is likely due to its diversity, where it has more examples so by chance some examples can fool our model versus MURA or MNIST that are more homogeneous datasets.

Finally, we observed that images from the Pneumonia dataset tended to be classified as outliers and we wondered if they were truly out of the training distribution or if it was caused by a failure of our outlier detection methods. We evaluated each of these images using the DenseNet-121 model and observed that the predictive performance increased when limiting which images to process using the ALI or AE SSIM score methods in Figure 6.

Given these results it is hard to conclude which model is better. An AE with SSIM score performs better at detecting OoD samples overall while an ALI model with SSIM score provides better isolates samples where the model will perform well. We use the ALI model because the reconstruction error is also informative.

## 4. Prediction explanation

We use the gradient saliency map discussed by Simonyan et al. (2014) and Lo et al. (2015) to explain why a network has made a prediction. Here for an input image $I$ and the pre-softmax output of a neural network $y$ we can compute the pixel-wise impact on a specific output $y_i$ or over all outputs. The computation cost is proportional to a feedforward pass of the network. The following saliency map is used for a general explanation of the prediction for task $i$: $\max(0, \partial y_i / \partial I)$. Figure 7 shows example saliency maps. Generally the gradient is high in clusters of pixels which cover regions which are predictive of the disease. One issue when interpreting gradients directly like this is that the gradient is high not only at the location of the feature (like a nodule) but also at locations which condition another region to have impact (like the area next to a nodule).

We compared this technique against another localization mapping called Class Activation Mapping (Zhou et al., 2015) (explained in Appendix §D). Figure 7 shows an example of localization of the Cardiomegaly disease using the two different mappings. The class activation mapping does not allow a very precise localization, as the resolution of the deep activation maps has been greatly reduced by successive pooling layers. Note that here the class activation map is upsampled to the size of the original image for visualization purpose.

## 5. Web Computation

The engineering of the system is designed in a modular way to be used by many projects in the future. We create a pipeline using ONNX to transform models trained in PyTorch or other frameworks. From ONNX we can transform the models into frameworks which have browser support such as TensorFlow or MXNet (Chen et al., 2015). For this project TensorFlow.js was the most compatible across browsers and supported the necessary operations to translate the DenseNet and ALI models from PyTorch. The conversion pipeline is as follows: PyTorch $\rightarrow$ ONNX $\rightarrow$ TensorFlow $\rightarrow$ TensorFlow.js.

The methodology when deploying to the browser is to package the model's graph and weights into files which are then loaded by a script running in the browser which will reconstruct the graph and load the weights. The script must process images into the format expected by the model, execute the computation graph, and then present the results. The source code is available here: REMOVED

We verified the computation graph conversion is correct by comparing predictions of the PyTorch model to predictions of the tensorflow.js model on three images. Differences between predictions were within a $1e^{-5}$ tolerance margin. Nonetheless, preprocessing methods differ, leading to different downscaled images. We validated on 20 images to assess prediction differences induced by the different preprocessing methods as shown in Appendix Figure E.5. In extreme cases, predictions differed by up to 20%, but the average difference is $\pm 3\%$, leading to consistent predictions between the two pipelines. This reveals that our model is somehow sensitive to aleatoric noise, a common issue in deep learning. More details about the web implementation can be found in Appendix §E.

## 6. Conclusion

In this work we present a complete tool to aid in diagnosing chest X-rays. We discuss the challenges of developing each aspect of the system and implement practical solutions in the areas of OoD detection, disease prediction, and prediction explanation. We believe that this is a solution to bridge the gap between the medical community and Deep Learning researchers.

### Acknowledgments

Acknowledgments withheld.

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

## Appendix A. Data

**Chest-Xray8 dataset** To train our models we used the Chest-Xray8 dataset released by the NIH (Wang et al., 2017). This dataset contains 108,948 frontal-view X-ray images of 32,717 unique patients with 14 disease image labels: Atelectasis, Cardiomegaly, Effusion, Infiltration, Mass, Nodule, Pneumonia, Pneumothorax, Consolidation, Edema, Emphysema, Fibrosis, Pleural Thickening, and Hernia.

There have been claims that this dataset contains many errors in labelling (Oakden-Rayner, 2017). However at the time of this project it was the only dataset of its size and usage rights.

**Pneumonia dataset:** For evaluation of out-of-distribution examples we would like the model to correctly predict a dataset from Guangzhou Women and Childrens Medical Center is used (Kermany et al., 2018). This dataset contains 5,232 chest X-ray images from children. 3,883 labelled as depicting pneumonia (2,538 bacterial and 1,345 viral) and 1,349 normal cases. This dataset was only used after all models had finished being trained to provide an unbiased evaluation.

**PadChest dataset:** The dataset (Bustos et al., 2019) contains 160,000 chest X-rays and reports gathered from a Spanish hospital spanning over 67,000 patients with multiple visits and views available. The number of labels is a superset of the labels of the labels in the NIH Chest-Xray8 dataset with 194 in total. We align these labels to the NIH Chest-Xray8 by matching the names exactly.

**MURA dataset:** For evaluation of out-of-distribution examples we would like the model to reject the MURA (musculoskeletal radiographs) dataset is used (Rajpurkar et al., 2018). It contains 40,561 bone X-ray images from 14,863 studies, where each study is manually labeled by radiologists as either normal or abnormal. The X-ray images contain a finger, wrist, elbow, forearm, hand, humerus, or shoulder.

We also experiment using the CIFAR-100 (Krizhevsky & Hinton, 2009) dataset of real world images as well as the MNIST (LeCun & Cortes, 1998) dataset of hand drawn digits.

## Appendix B. Model Training

Here are extra details of the training which are removed from the main text.

We use Adam with standard parameters ($\beta_1 = 0.9$ and $\beta_2 = 0.999$), learning rate of 0.001, and learning rate decay of 0.1 when validation accuracy plateaus.

## Appendix C. ALI Training

An overview of the ALI model (Dumoulin et al., 2016) is shown in Figure 4. Images were resized to $64 \times 64$ pixels. 106,381 images were used for training, 1000 images were used for validation for the ALI model. When training the ALI model most hyperparameters resulted in the same performance. We tried latent space sizes of 64, 128, 256, and 512 and learning rates of 0.000001 and 0.00001. To stabilize training we used two tricks, first we halt training of the generator when its performance becomes too good. Also, label smoothing is used where true positive labels are modified to between 0.7 and 1.1 and true negative labels between -0.1 and 0.3 (Salimans et al., 2016). We used an Adam optimizer with $\beta_1 = 0.5$ and

$\beta_2 = 1e^{-3}$. Each ALI model during the hyperparameter search was trained for 2,121,000 iterations or until the server crashed using a batch size 100.

## Appendix D. Class Activation Mapping

Class Activation Mapping (Zhou et al., 2015). Let $f_k(x, y)$ be the activation map of the $k^{th}$ filter of the last convolutional layer at location $(x, y)$. A global pooling operation is performed on this activation map before being fed to the last FC layer. Thus, let $\omega_{ck}$ be the weight of the last FC layer that links filter $k$ to class $c$. The Class Activation Map for class $c$ at location $(x, y)$ is computed as: $M_c(x, y) = \sum_k \omega_{ck} f_k(x, y)$. This captures the part of the activation map after the last convolutional layer that has a strong impact on the prediction of class $c$.

## Appendix E. Extra Implementation Details

**Other design considerations**   To ensure accurate image representation we used canvas elements instead of creating image objects. This offers exact control over the pixels of an image. When performing image preprocessing we scale and crop instead of stretching images. This is to ensure that structure remains correct.

**Runtime**   There are three main processing points: initial loading of the models ($12 \pm 2$ seconds), computing the ALI and DenseNet computation graphs ($1.3 \pm 0.5$ seconds), and computing gradients to explain predictions ($17 \pm 4$ seconds).

There were two ways we can achieve gradient computations in the browser. We used the `tf.grad` method of TensorFlow.js which has the advantage that it can work on any model loaded into the site while having the disadvantage of slower running time. An alternative is to build a gradient computation graph and export that graph as regular model. This has the advantage that the graph could be built offline and optimized instead of being built in realtime in the browser. However there seem to be limitations in the ability to export the computation graphs of the gradients due to missing backward operations and gradient computation graphs that are not compatible with ONNX.

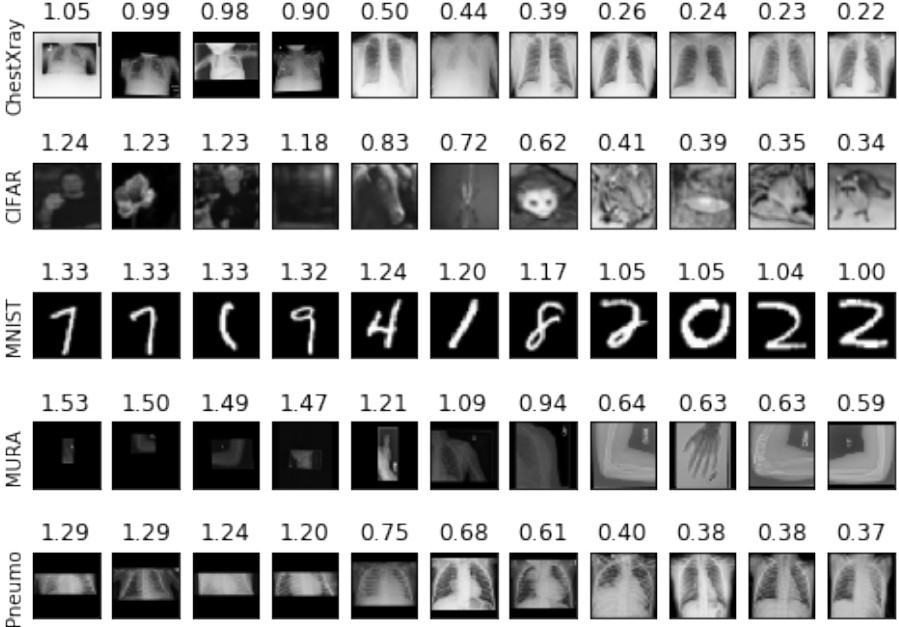

(*a*) Sample images using the L2 Reconstruction Loss (lower score is better)

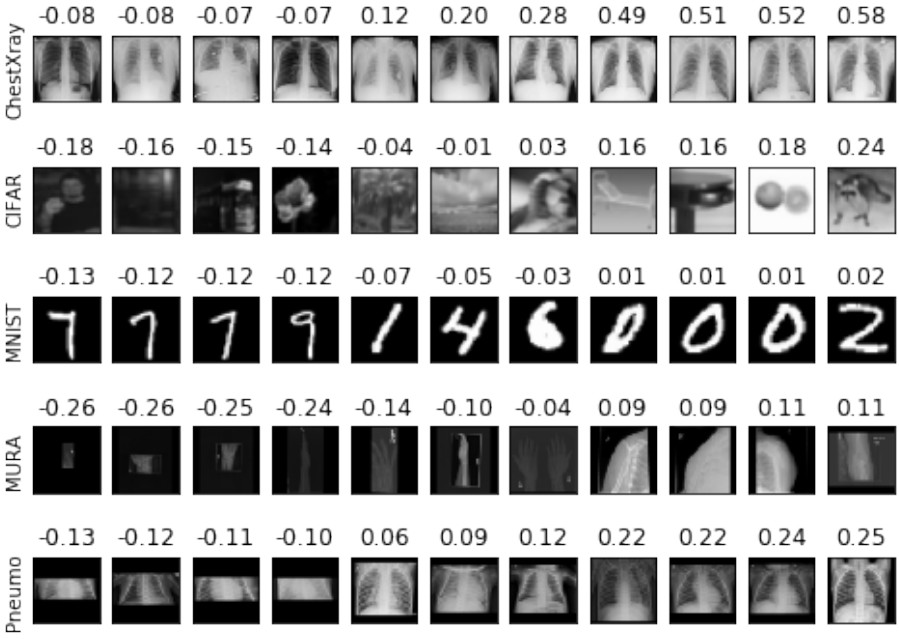

(*b*) Sample images using SSIM (higher score is better)

Figure E.1: Reconstruction error shown over different datasets. The model is not able to reconstruct X-rays other than chest X-rays so the reconstruction error is high. For each dataset images to the left are harder to reconstruct and images to the right are easier.

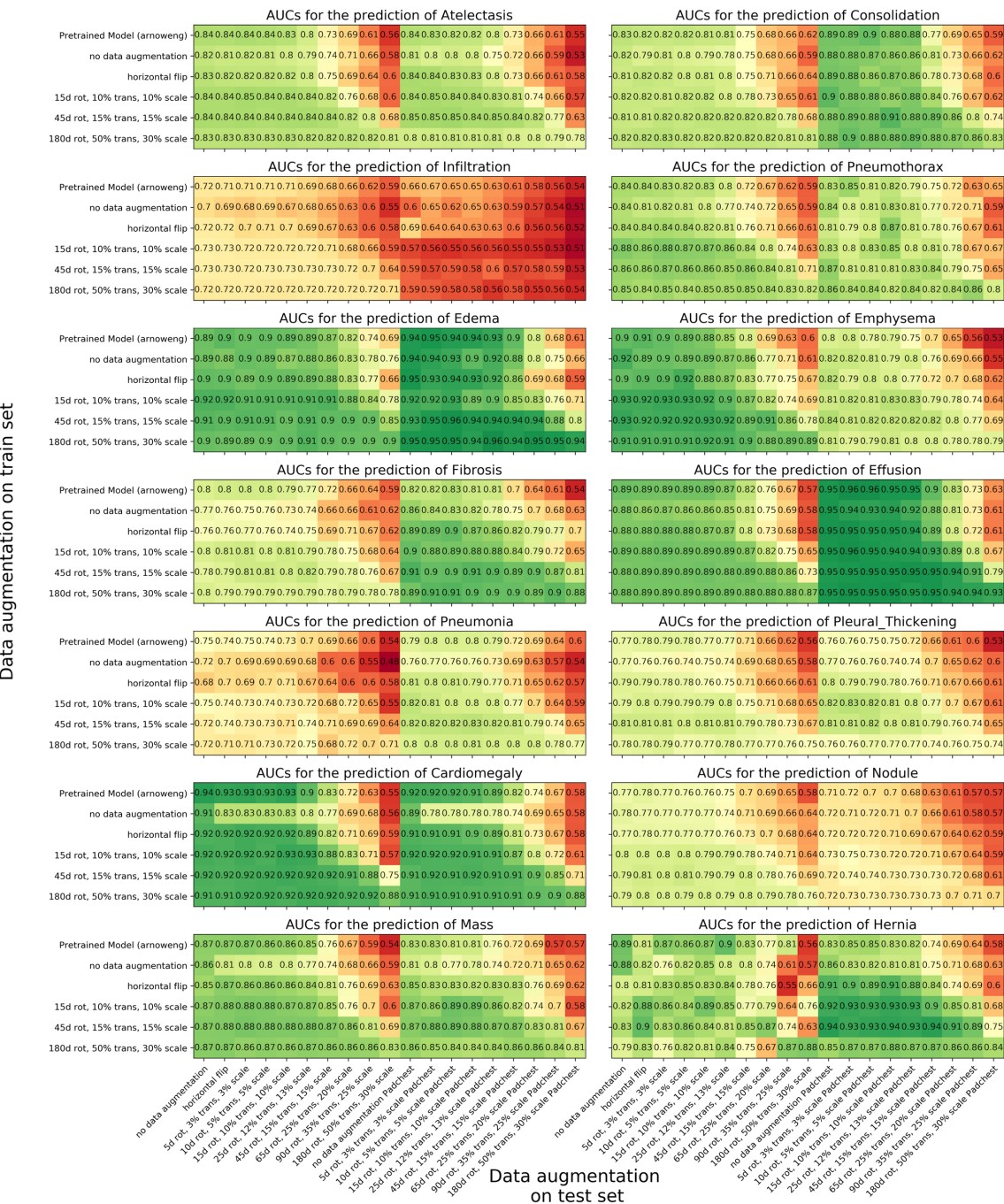

Figure E.2: Robustness of the different models to data augmentation on the test set of the ChestXray dataset (left half of matrices) and on the PadChest dataset (Bustos et al., 2019) (right half of matrices). There is an increasing level of data augmentation at train time (from top to bottom) and at test time (from left to right).

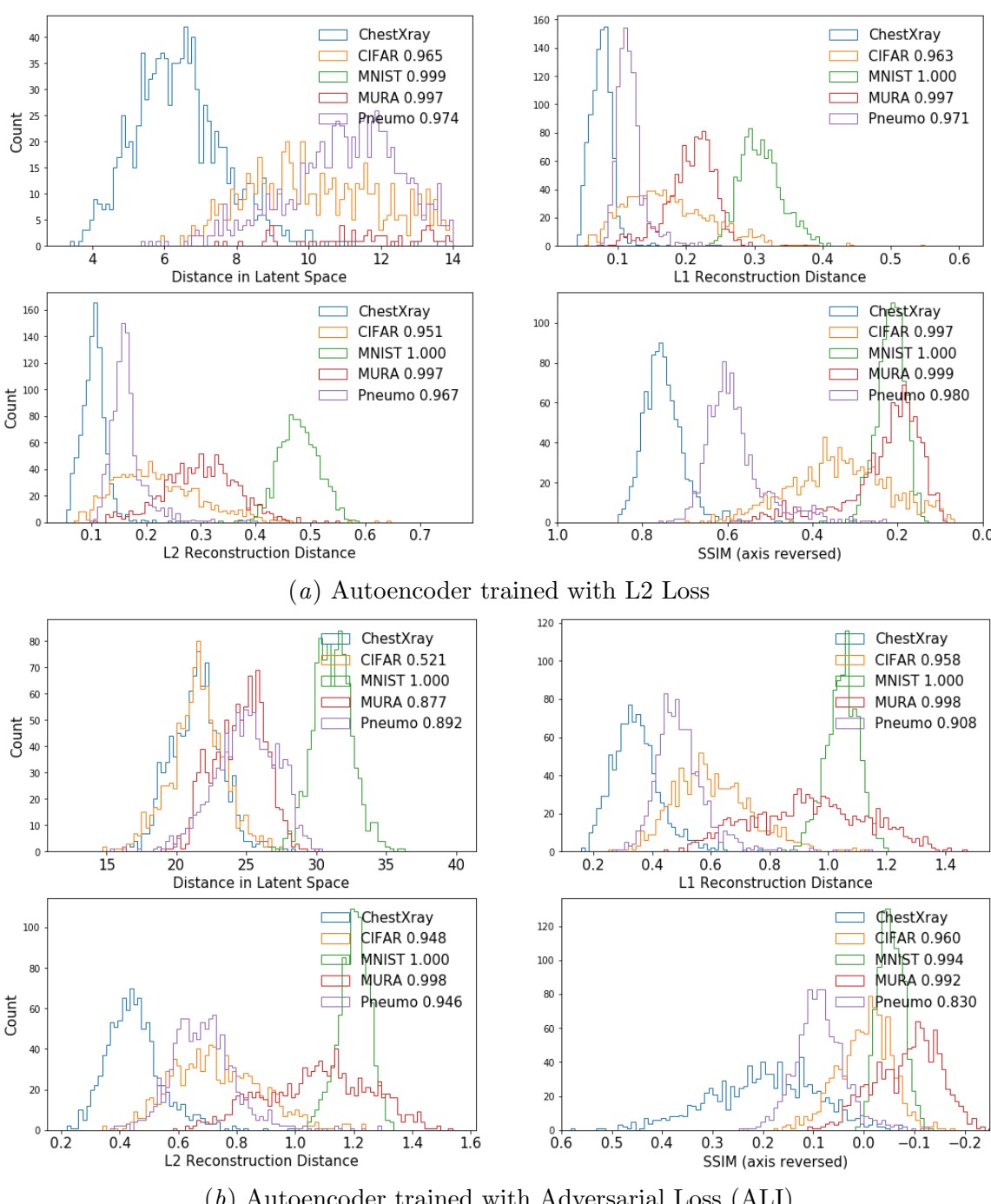

(*a*) Autoencoder trained with L2 Loss

(*b*) Autoencoder trained with Adversarial Loss (ALI)

Figure E.3: Distribution of OoD metrics using different metrics. Euclidean distance in the latent space and three types of pixel-wise reconstruction losses: L1, L2, and SSIM. The AUC when separating the ChestXray from each other dataset is shown in the legend.

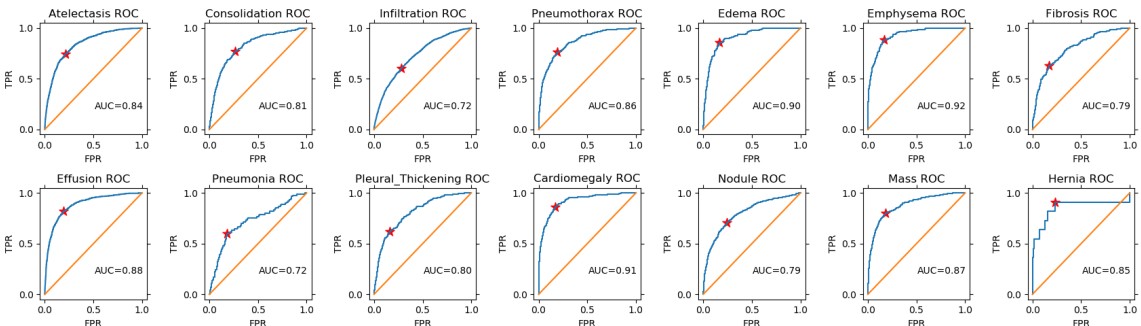

Figure E.4: ROC of 14 diseases on the NIH ChestXRay holdout set. The red star corresponds to the operating point. We decide on an operating point and scale the output of the model to reflect this performance.

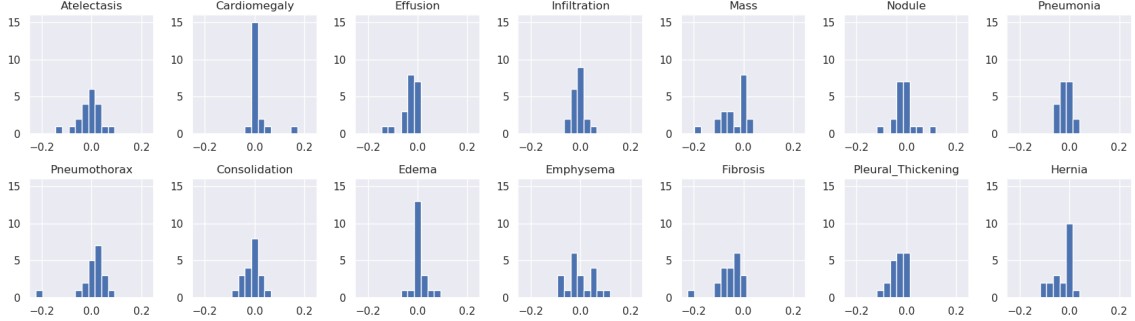

Figure E.5: Histogram of differences between predictions using the JavaScript/WebBrowser image preprocessing pipeline and using the PyTorch pipeline, for each disease. This validation is performed over 20 randomly chosen images.

