# OpenReview forum: "Chester: A Web Delivered Locally Computed Chest X-Ray Disease Prediction System"
_MIDL.io/2020/Conference — Submitted to MIDL 2020_

### Official Review · AnonReviewer3 · 2020-03-12
**An Interesting system needs further development**

**Rating:** 3
**Confidence:** 4

**Summary:**

This work tries to bridge the gap between image classification with deep learning in the medical field of chest X-ray diagnosis and medical practice. The authors build a web-based locally-running system to help the classification and its explanation in lung disease diagnosis. Their work is significant in helping medical specialists in such a diagnosis scenario with high system adaptivity which reduces the labor and increases trust.

**Strengths:**

1. A web-based system that allows applications in different system environments.
2. Thoroughly validation in the method.
3. A strict criterion to avoid outlier inputs that affect the prediction of the system.


**Weaknesses:**

1. Although outlier extraction/prevention is very useful to avoid incorrect predictions, there is no record on how much rate of the outliers is in a practical scenario. If the rate is moderate or high which might be true since different protocols are applied in different centers, the application would be very limited. For example, the Pneumonia dataset shows quite a different distribution in Figure 5 but they are also chest x-ray images.
2. For these outliers detected, there is no further heuristics/methods for chest x-ray images.


**Detailed Comments:**

1. Figure 5 caption. It mentions about L1 and latent variable distance but not a match to the figure, please fix.
2. In Figure 1, what is the reconstruction score? If it means the OOD score, which one is applied,. L1, L2 or latent variable distance? Need to clarify. Meanwhile, what does this score serve here? The doctor would not need extra info if it is not helpful or clearly defined.
3. In Figure 1, what is the OOD heatmap? If it means the pixel-wise difference is it necessary for doctors?


**Justification Of Rating:**

This work allows a simple deep-learning classification network applied to chest x-ray images. It would definitely be of great favor for medical doctors during diagnosis. However, the application scenarios is limited by its stringent outlier detection. The system should also provide a method/heuristics to address problems when a chest x-ray is classified as an outlier.

**Paper Type:**

validation/application paper

**Special Issue:**

no

---

> ### Author Response · Authors · 2020-03-26
> **Author response**
>
> We thank the reviewer for their feedback!
>
> > Although outlier extraction/prevention is very useful to avoid incorrect predictions, there is no record on how much rate of the outliers is in a practical scenario. If the rate is moderate or high which might be true since different protocols are applied in different centers, the application would be very limited
>
> There is a trade off between the acceptance rate of the OOD detector and the performance of the predictions. The threshold could be easily changed in the code, and it would be feasible to change it according to the needs of physicians if they were willing to accept a higher error rate.
>
> > For these outliers detected, there is no further heuristics/methods for chest x-ray images.
>
> This is by design. We choose to reject these samples without any further interpretation in order to avoid providing potentially misleading predictions outside of the training distribution. To solve this we should train on more diverse data.
>
> > For example, the Pneumonia dataset shows quite a different distribution in Figure 5 but they are also chest x-ray images.
>
> This was not clear in the paper. We made the text more clear to indicate that the Pneumonia dataset was pediatric X-rays which our model was not trained on. We added "As we did not train on pediatric cases we do not expect this dataset to be fully in distribution but we expect it to be separated from the OoD datasets (CIFAR, MNIST, and MURA)."
>
> We are running experiments now to add more chest X-ray datasets which should be in-distribution to Figure 5 which should address this concern. We will respond to your review with the new figure. (edit: here is the new Figure 5 showing the scores for the PADCHEST dataset: https://i.imgur.com/C15g3sN.jpg )
>
> > 1. Figure 5 caption. It mentions about L1 and latent variable distance but not a match to the figure, please fix.
>
> Fixed.
>
> > 2. In Figure 1, what is the reconstruction score? If it means the OOD score, which one is applied,. L1, L2 or latent variable distance? Need to clarify. Meanwhile, what does this score serve here? The doctor would not need extra info if it is not helpful or clearly defined.
>
> Fixed. This score would be used by the developer of a system to calibrate the needed performance. As shown in Figure 6 a lower threshold provides higher performance. I think a doctor should not have to think about this unless the system prevented them from processing an image (like the one proposed) which would force them to crop the image and try again or make a decision without the tool.
>
> > 3. In Figure 1, what is the OOD heatmap? If it means the pixel-wise difference is it necessary for doctors?
>
> It is the pixel-wise difference. As people often ask "Why isn't my X-ray working?" they can use the OoD heatmap to understand why. We have added a qualitative analysis of how useful the methods are in figure A.6 in the appendix which you can see here: https://i.imgur.com/BKW3C6G.jpg

---

> > ### Comment · AnonReviewer3 · 2020-04-02
> > **Thanks for the response**
> >
> > I find this work very promising. However, at this stage of design, I am not convinced to change my review. Hopefully, a better solution to the outliers will come along with this project.

---

### Official Review · AnonReviewer1 · 2020-03-12
**Authors present a web based interface that can be used to aid doctors in identifying irregularities in chest x-rays or stand as a sort of second opinion.**

**Rating:** 4
**Confidence:** 4
**Recommendation:** Oral

**Summary:**

In this paper, authors describe a deep learning model, built with a DenseNet-121 architecture, to process chest x-rays and identify possible risks based on them. a Authors present a web based interface that can be used to aid doctors in identifying irregularities in chest x-rays or stand as a sort of second opinion.

**Strengths:**

Both the paper structure and the prototype are clear. The web interface for the Chester prototype appears to be a very easy to use tool. I like that it identifies the image regions that influence the prediction, allowing for doctors to better understand what is happening within the system. I also like the visual bars of healthy vs. risk predictions.

Authors appeared to go to appropriate lengths to train and test their model. They expanded their dataset by augmenting images through image rotations, scaling, and translations. These augmentations did not seem to harm their model’s performance.

Another strength of this paper is the authors’ focus on patient privacy and discussion of the goals of this tool. I also appreciate that authors specify that this tool is intended to complement the skills of students, doctors, or radiologists. They also discuss challenges and possible solutions for those challenges. The authors seemed to have an answer to why they did pretty much everything.


**Weaknesses:**

This is a very strong paper.  There are no obvious weaknesses.
As it is an application like paper, this warrants a paper more focused on results.  This was the case in this paper and the authors do provide sufficient results

**Detailed Comments:**

see above

**Justification Of Rating:**

This is a strong and well written paper.  The authors justify their approach and it has no obvious weaknesses.
Authors appeared to go to appropriate lengths to train and test their model.
Both the paper structure and the prototype are clear

**Paper Type:**

validation/application paper

**Questions To Address In The Rebuttal:**

See above.  Strong accept.
While the out of sample detector is a good idea, it potentially doesn't account for in sample - yet out of training distribution errors.

**Special Issue:**

yes

---

> ### Author Response · Authors · 2020-03-26
> **Author response**
>
> We thank the reviewer for their feedback!

---

### Official Review · AnonReviewer4 · 2020-03-13
**Nicely written paper describing a web-based tool for chest X-ray diagnostics**

**Rating:** 3
**Confidence:** 4
**Recommendation:** Poster

**Summary:**

The authors present a software tool which implements the automatic scoring of chest X-rays based on a public dataset. The code will be made available by the authors. Their model can run in a browser using Tensorflow.js (client-side). To achieve this, the authors took their models trained in pytorch, and passed these through ONNX to Tensorflow and Tensorflow.js. Additionally, the authors provide a way of detecting out-of-distribution models. This can be useful when the radiologist uploads images of his cat, or when the X-ray can not be judged properly.

**Strengths:**

- Well written. The paper is clearly about an engineering problem as there are no new methods developed. However, the line of thinking is present and easy to follow.
- Code available, using public dataset
- Adding an out of distribution detector
- Thorough evaluation of the classification model

**Weaknesses:**

- As there are essentially no new methods developed, it would be worthwhile to see how well this out-of-distribution detector actually works. Are there surely no X-rays which can be properly read by radiologist, yet get rejected by the model? The evaluation seems thorough, but is not convincing. The different metrics chosen seem to be a bit ad hoc.


**Justification Of Rating:**

The paper provides no new insights into the problem of Chest X-ray classification. This was also not the intention of the authors, and they wanted to develop this as a tool to show the possibilities of deep learning and to build a prototype which can be easily used by many. I agree they have succeeded in this. However, the models themselves are out-of-the-box and the evaluation for the out-of-distribution detection is not convincing. It is however a complete prototype, and deserves to be shown.

**Paper Type:**

validation/application paper

**Special Issue:**

no

---

> ### Author Response · Authors · 2020-03-26
> **Author response**
>
> We thank the reviewer for their feedback!
>
> > The paper provides no new insights into the problem of Chest X-ray classification.
>
> We provide insight into the challenges of deploying models. The limitations of Out of Distribution detection that we encountered should motivate the community to focus their attention on improving the performance and usability of these methods. Also, the manner in which we overcome these limitations will be of great interest to any deep learning practitioner who seeks to deploy their model.
>
> Moreover, we performed a systematic study of the effect of data augmentation on the performance of the model in the context of Chest X-ray images. Our results reveal how much data augmentation the DenseNet model can bear in that specific context, which is of interest to anyone working with chest X-ray images. Also, these results highlight the effect of data augmentation on model generalization, which should be of interest to researchers studying model generalization.
>
> Finally, our project is open source to enable researchers in the MIDL community to deploy their models in a similar way. We hope that this code becomes the goto for deploying models quickly without costly overhead and privacy concerns.
>
>
> >  it would be worthwhile to see how well this out-of-distribution detector actually works.
>
> We only focused our evaluation on reconstruction based approaches. Some reasons are classifier based approaches like ODIN are not applicable to our multi-task setting. Also, methods like MC-dropout are too computationally expensive for our use case.
>
> We performed an analysis with multiple methods and datasets. We made the text more clear to indicate that the Pneumonia dataset was pediatric X-rays which our model was not trained on.
>
> > Are there surely no X-rays which can be properly read by radiologist, yet get rejected by the model?
>
> Yes there are, imagine an X-ray rotated 90 degrees. A radiologist can read this but it would be rejected by our model.
>
> As we showed in Figure 6 we reject samples of pediatric X-rays that the model would perform poorly on. Given that they had labels in that dataset a radiologist trained in this type of case would be able to read these X-rays but we show that the model cannot perform this task well.
>
> The purpose of the Out of Distribution pipeline is to ensure that the model does not classify examples that lie outside of its training distribution, which is not meant to encompass everything that a radiologist can read.
>
> > The evaluation seems thorough, but is not convincing.
>
> How can we improve the evaluation?

---

### Official Review · AnonReviewer2 · 2020-03-15
**Not suitable for MIDL**

**Rating:** 1
**Confidence:** 4

**Summary:**

This paper has demonstrated a web-delivered deep learning tool for chest x-ray analysis. The authors have implemented/used prior deep learning methods for disease prediction, outlier detection, and prediction detection. They claim that this work can "bridge the gap between the medical community and deep learning researchers", which is not shown in the results. This work could bring interesting discussions between radiologists/physicians and deep learning researchers, but less than 1% of the MIDL attendees are radiologists/physicians.

**Strengths:**

This work is a promising application that can potentially help radiologists and physicians utilize deep learning techniques for chest x-ray analysis in their clinical practice. The authors made their source code public.

**Weaknesses:**

This paper is lack of experimental results to suggest that "this is a solution to bridge the gap between the medical community and deep learning researchers". It is very hard to imagine how a radiologist or a physician would use this tool to read an image from their PACS system. The audience of this paper should be radiologists, who are not likely to go to MIDL.

**Justification Of Rating:**

The authors have used off-the-shelf deep learning methods to develop a web-delivered tool for radiologists and physicians. The authors claim that "this is a solution to bridge the gap between the medical community and deep learning researchers" but haven't shown any experimental results (e.g. user studies) to support this. This work will not bring interesting discussions to the MIDL community, most of who are "deep learning researchers". The authors should consider submitting this to a conference like RSNA.

**Paper Type:**

validation/application paper

**Special Issue:**

no

---

> ### Author Response · Authors · 2020-03-26
> **Author response**
>
> We thank the reviewer for providing honest and critical feedback.
>
> > This paper is lack of experimental results to suggest that "this is a solution to bridge the gap between the medical community and deep learning researchers".
>
> Our evidence for this is that the tool is accessible for free and can be used on a laptop or desktop of a radiologist/physician. It is also open source to enable other deep learning research groups to build similar tools that connect with radiologists/physicians.
>
> We have to start somewhere. The first step in this process is to overcome the challenges of building such a system and validating each component. Having the deep learning research community stand behind that evaluation will allow radiologists/physicians to feel confident that this corresponds to what can be done with modern deep learning methods.
>
> > It is very hard to imagine how a radiologist or a physician would use this tool to read an image from their PACS system.
>
> PAC systems can export image files (png, jpeg). These can be processed with the tool in a browser on the same machine. We have done it many times without issue. We also don't expect this to be part of the everyday pipeline because it is not made for medical use.
>
> > This work will not bring interesting discussions to the MIDL community, most of who are "deep learning researchers".
>
> We provide insight into the challenges of deploying models. The limitations of Out of Distribution detection that we encountered should motivate the community to focus their attention on improving the performance and usability of these methods. Also, the manner in which we overcome these limitations will be of great interest to any deep learning practitioner who seeks to deploy their model.
> Moreover, we performed a systematic study of the effect of data augmentation on the performance of the model in the context of Chest X-ray images. Our results reveal how much data augmentation the DenseNet model can bear in that specific context, which is of interest to anyone working with chest X-ray images. Also, these results highlight the effect of data augmentation on model generalization, which should be of interest to researchers studying model generalization.
>
> Finally, our project is open source to enable researchers in the MIDL community to deploy their models in a similar way. We hope that this code becomes the goto for deploying models quickly without costly overhead and privacy concerns.
>
> > less than 1% of the MIDL attendees are radiologists/physicians.
>
> Even if this were true. The community can grow with the help of this tool to enable more radiologists/physicians to experiment with these tools and start projects which will lead them to publish at MIDL. Moreover, the paper is interesting for the Deep Learning Community as emphasized in the previous response.
>
> > The authors have used off-the-shelf deep learning methods to develop a web-delivered tool for radiologists and physicians.
>
> These components have never been combined like this into a complete tool.
>
> > The authors should consider submitting this to a conference like RSNA.
>
> We did submit to RSNA2019 and the paper was not accepted and without feedback.

---

> > ### Comment · AnonReviewer2 · 2020-04-01
> > **The work is valuable**
> >
> > Thank you for your response. There's no doubt that this work and the authors' efforts should be appreciated. I understand the challenge of finding a suitable venue for publishing this type of interdisciplinary work.

---

### Meta-Review · Area_Chair1 · 2020-04-06
**MetaReview of Paper52 by AreaChair1**

**Rating:** 3
**Recommendation For Accepted Papers:** Poster

**Metareview:**

This paper presents a web-based tool for chest x-ray diagnosis. The paper is interesting from the point of view of illustrating and discussing with the MIDL community what it will take for our methods to be widely and freely applicable to practitioners, beyond our own collaborating clinicians.

**Paper Type:**

validation/application paper

**Special Issue:**

no

---

### Decision · Program_Chairs · 2020-04-11

**Decision:**

Reject

**Comment:**

Based on the reviews and AC comments, the paper has merit leading to an AC borderline accept recommendation. Given the large number of excellent submissions received, unfortunately, the paper did not meet the conference acceptance criterion.